# Feasibility of Hyperspectral Single Photon Lidar for Robust Autonomous Vehicle Perception

**DOI:** 10.3390/s22155759

**Published:** 2022-08-02

**Authors:** Josef Taher, Teemu Hakala, Anttoni Jaakkola, Heikki Hyyti, Antero Kukko, Petri Manninen, Jyri Maanpää, Juha Hyyppä

**Affiliations:** 1Department of Remote Sensing and Photogrammetry, Finnish Geospatial Research Institute FGI, National Land Survey of Finland, 02150 Espoo, Finland; teemu.hakala@nls.fi (T.H.); anttoni.jaakkola@gmail.com (A.J.); heikki.hyyti@nls.fi (H.H.); antero.kukko@nls.fi (A.K.); petri.manninen@nls.fi (P.M.); jyri.maanpaa@nls.fi (J.M.); juha.coelasr@gmail.com (J.H.); 2Department of Computer Science, Aalto University School of Science, 02150 Espoo, Finland

**Keywords:** single photon, hyperspectral LIDAR, classification, autonomous driving, object detection, SPAD, remote sensing, multispectral, spectral signature, photon shot noise

## Abstract

Autonomous vehicle perception systems typically rely on single-wavelength lidar sensors to obtain three-dimensional information about the road environment. In contrast to cameras, lidars are unaffected by challenging illumination conditions, such as low light during night-time and various bidirectional effects changing the return reflectance. However, as many commercial lidars operate on a monochromatic basis, the ability to distinguish objects based on material spectral properties is limited. In this work, we describe the prototype hardware for a hyperspectral single photon lidar and demonstrate the feasibility of its use in an autonomous-driving-related object classification task. We also introduce a simple statistical model for estimating the reflectance measurement accuracy of single photon sensitive lidar devices. The single photon receiver frame was used to receive 30 12.3 nm spectral channels in the spectral band 1200–1570 nm, with a maximum channel-wise intensity of 32 photons. A varying number of frames were used to accumulate the signal photon count. Multiple objects covering 10 different categories of road environment, such as car, dry asphalt, gravel road, snowy asphalt, wet asphalt, wall, granite, grass, moss, and spruce tree, were included in the experiments. We test the influence of the number of spectral channels and the number of frames on the classification accuracy with random forest classifier and find that the spectral information increases the classification accuracy in the high-photon flux regime from 50% to 94% with 2 channels and 30 channels, respectively. In the low-photon flux regime, the classification accuracy increases from 30% to 38% with 2 channels and 6 channels, respectively. Additionally, we visualize the data with the t-SNE algorithm and show that the photon shot noise in the single photon sensitive hyperspectral data contributes the most to the separability of material specific spectral signatures. The results of this study provide support for the use of hyperspectral single photon lidar data on more advanced object detection and classification methods, and motivates the development of advanced single photon sensitive hyperspectral lidar devices for use in autonomous vehicles and in robotics.

## 1. Introduction

Autonomous driving is expected to be one societally significant reform coming to modern society during the next 20 years. In some estimates, up to 15% of new cars will be completely autonomous in some US city areas in early 2030. Based on US studies, the further impacts of autonomous driving include improvement of fuel economy [1], platoon driving that would save additionally 20–30% fuel consumption [2], productivity gains while commuting [3], stress reduction [4], and decline of required parking spaces [3]. It has been estimated that due to the changes brought about by autonomous cars, 39% of urban space will become available to new sorts of use. According to a study by [5], autonomous vehicles would on average lead to a 38-h reduction in commuting time per individual per year, as well as saving the US economy alone USD 1.3 trillion per year.

Autonomous vehicles often include a large number of sensors mounted on-board due to the need to observe the environment all around the vehicle and also to position the vehicle. The perception sensors include ranging sensors and vision-based sensors. Ranging sensors, such as lidar, radar, and sonar, provide 3D measurements of the surroundings [6]. They are also active sensors providing their own energy for target illumination. The advantages of lidar include high pulse repetition rate, high accuracy of range measurements, and small beam divergence, allowing the separation of small objects from each other. Micro- and millimetre wave radars are feasible for range, distance, and speed measurement, but, due to a wider beam, the separation of objects is weaker than with lidar. Sonar are mainly limited to short ranges around the car. Passive-vision-based sensors collect both radiometry and geometric information of the surroundings. Cameras and videos are feasible for discriminating traffic lights, signs, and fast-moving objects. Thermal images can be used to discern warm objects (animal and people) from cold ones. In general, objects are detected using the following properties: geometry, spectral response, bidirectional information, and time series information. Single shot laser scanning typically only uses geometry and geometrical features derived from point clouds and, therefore, automatic classification of an object is expected to be improved by increasing spectral properties of the target.

Even though multi-sensor approaches are used for road environment target detection, classification, and tracking, there are several shortcomings with the current autonomous sensors: (1) in some use cases it is necessary to see 200–300 m ahead and be able to detect the objects, (2) current systems provide non-optimal accuracy of main road-user classification, (3) energy usage of the future cars using current state-of-art sensors is too high, and (4) sensor technology should work in all weather and climate conditions. In order to solve these problems, the sensor technology is leading to an integrated lidar and camera solution. Multispectral or hyperspectral, single-photon, possibly frame-based solid-state, lidar is one potential candidate technology for the future.

In a paper [7], the concept of using single-photon lidar for autonomous vehicles with its principles, challenges, and recent advances was presented. The long-range capacity, high depth resolution, and use of eye-safe laser sources are key drivers of the single-photon lidar towards autonomous driving. Additionally, a growing research area is non-line-of-sight (NLOS) imaging in which diffuse surfaces, such as roads or walls, could act as mirrors, allowing vision around obstacles [7]. There are already some early works for developing single photon lidars for autonomous driving [8,9,10]. According to [11], single photon techniques are already applied by Ouster OS-1 64, Toyota having developed a single-photon avalanche diode (SPAD) with enhanced near-infrared (NIR) sensitivity for use in future automotive light detection and ranging (LIDAR) systems [12], and Princeton Lightwave (acquired by Argo.ai) by having realized a SPAD lidar prototype.

In this paper, we describe the prototype hardware for hyperspectral single photon solid-state lidar, introduce a statistical model for estimating the spectral reflectance measurement accuracy in a low-photon flux regime (less than $10^{2}$ detected photons per wavelength channel), and perform a feasibility study of using hyperspectral single photon receiver in autonomous-driving-related object classification task. In our study, the dimensions of the receiver frame (32 × 32 pixels receiver obtained from Princeton Lightwave) are used to record the intensity as a function of wavelength.

## 2. Related Work

Use of multispectral lidar for autonomous ground vehicles has already been proposed [13]. They presented a supercontinuum-based (25 spectral bands between 1080–1620 nm) multispectral lidar concept for military applications in order to identify objects based on combined spatial and spectral features, up to a maximum of 150 m in distance. Hyperspectral lidars are typically accomplished, employing supercontinuum laser sources. Some of the technologies to generate supercontinuum laser sources, i.e., “white lasers” can be read from [14,15]. The early concept describing hyperspectral lidar in object classification can be found in [16] and early prototypes in [17,18]. Since then, hyperspectral lidar has been used for various applications studies, such as classification of spruce and pine trees [19], estimation of rice leaf nitrogen content [20], leaf chlorophyll estimation [21,22,23], architecture preservation [24], ore classification [25], target detection over time series [26], automated point cloud classification and segmentation [27,28], and vegetation red edge parameters extraction [29]. Multispectral lidar can also be accomplished by employing several monochromatic laser sources or by having a lidar transmitter transmitting several wavelengths. In [30], a multispectral integrated detector array, including detectors capable of detecting the range and spectral components, were reported for the first time.

Single-photon timing has also emerged as a candidate technology for high-resolution 3D imaging. In [31], the first fully integrated frame system for single-photon time-of-arrival evaluation was performed. In addition to the high timing resolution (some picoseconds), single-photon detectors lead to detection over longer ranges and/or allows the use of lower power laser sources. In [11], long-range single-photon 3D imaging was demonstrated with target distance up to 45 km along the Earth’s atmosphere at sea level, and in [32], over a distance of 201.5 km at an elevation of 1770 m. In [33], the long-range, low-power capability of a single-photon lidar was demonstrated by measuring targets at an average optical power of 10 mW, while achieving a maximum measurement range of 10 km. In [34], they presented an optical 3D ranging camera for automotive applications that is able to provide a centimetre depth resolution over a $40^{{}^{\circ}}\;\times \;20^{{}^{\circ}}$ field of view up to 45 m at 808 nm. Single-photon approaches have been used for demonstrations of multispectral depth imaging for target identification [35,36], quantification [37], land cover classification [38,39] and for the measurement of the physiological parameters of foliage [40].

Conventional lidar employs a narrow beam with a scanning mechanism. Alternatively, an object can be illuminated by using a wide-field flash and receiving backscattered returns using frame-based imaging. The 3D flash lidar has a number of advantages over conventional point (single pixel) laser scanners. In contrast to laser scanning systems, no mechanical moving parts are needed. Flash lidar is also capable of composing a 3D image of a scene in just one shot. As the cameras reach up to several hundreds frames per second, they are ideally suited to be used in real-time applications. Other flash lidar advantages include lightweight, blur-free images without motion distortion, no need for precision scanning mechanisms, thus, no moving parts are needed, and ability to discriminate distributed targets through range-gating. Ref. [41] present the idea of multispectral frame-based lidar. The detector arrays were made of uncooled InGaAs device for 1.5 µm wavelength and of cooled HgCdTe device for 3.8 µm wavelength.

Various measurement models have been proposed for extracting the return pulse signal or reconstructing the 3D scene from noisy, low-photon flux, single-photon sensitive measurements, for example, in a multiple target scenario [42], and when multispectral information is available [43]. Although the results are often good, the optimization step increases latency for real-time operation. In order to address the issue, Ref. [44] has proposed the use of plug-and-play point cloud denoising tools for low-latency data processing.

Multispectral lidars typically implement return pulse separation into wavelength channels by passing the incident photons into a spectrograph in the receiver side [18,40,45]. The spectral separation can be employed also at the transmission side. In [46], an acousto-optic tunable filter (AOTF) was added to the transmission side to select the desired wavelength pattern from consecutive supercontinuum laser illumination pulses. Spectral separation is possible, not only in the spatial-dimension, but also in the time-dimension. In [47], the chromatic group delay dispersion properties of the supercontinuum lasers non-linear optical fibre were used to perform wavelength-to-time mapping which allowed the spectral information to be resolved via an additional measurement model optimization step. Other approaches include, for example, Ref. [48] where the use of plasmonic colour filter attached in front of a SPAD array has been demonstrated in a single-photon multispectral fluorescence imaging study.

Recently, large form factor SPAD arrays have become more common. A 512 × 512 pixel SPAD array with range gating possibility was introduced in [49]. In [50], a SPAD array with megapixel resolution (1024 × 1000 pixels) and high photon detection efficiency was demonstrated in a frame-based range-gated lidar experiment. The recent upward trend in the resolution of SPAD arrays increases the attractiveness of using the spatial wavelength routing approach, where a spectrograph is used at the receiver side, for realizing a single-shot frame based hyperspectral lidar.

## 3. Materials and Methods

### 3.1. The Prototype Hyperspectral Single Photon Lidar and Its Operating Principle

The applied detector was a 32 × 32 SPAD array, Kestrel (Princeton Lightwave). Each element has a TDC (Time to Digital Converter) with timing resolution of 250–1250 ps and measures the time between the initialization of the sensor and the first detected photon hitting the element. The sensor is initialised by a trigger pulse from the supercontinuum laser source (Leukos Samba 400) when a laser pulse is generated. After a set time period of 2–10 μs (dependent on the timing resolution) the detector outputs an image, where, instead of intensity, each element represents the timer value. The detector is sensitive to light at a spectral range of 920–1620 nm, and can acquire frames at a rate of 186 kHz. The laser pulse repetition rate is 30 kHz.

The laser pulse was collimated by a refractive collimator and transmitted to a target using an adjustment mirror for beam alignment (Figure 1). The scattered light from the target was collected using a 3 inch diameter off-axis parabolic mirror and focused to an optical fibre. A mirror mounted to a gimbal mount was used to align the field of view of the fibre to the footprint of the laser. The fibre holder has a fine adjustment screw for focus. The off-axis parabolic mirror has a hole in the middle for the transmitted laser pulse to pass through. There is also a place for optional filters and a photodiode for triggering external equipment from the laser pulse.

The other end of the fibre was connected to detection optics, where the light was collimated with a parabolic mirror to a diffraction grating with 150 lines/mm (Figure 2). An achromatic lens was used to refocus the light to the detector. A cylindrical lens was placed in front of the detector to distribute the light over the detector area in the direction opposite to the spectral distribution. Therefore, each column of elements of the sensor receives light at the same wavelength (Figure 3). For each of the 32 wavelength bands, the sensor has 32 elements to determine the intensity.

Beam scanning can be realized on the prototype hyperspectral single photon lidar by employing standard beam scanning mechanisms, such as rotating multi-faceted, Palmer scanning, or oscillating mirrors. Additionally, the optical head assembly can be designed to be operated on a rotating assembly.

### 3.2. Our Statistical Model for Spectral Reflectance Measurement Accuracy in the Low-Photon Flux Regime

A fundamental statistical property of light incident on a detector element is that the arrival times of individual photons follow the Poisson distribution. This property can be derived both from the semi-classical and the quantum mechanical model of coherent light [51]. The desire with single photon sensitive lidars is to detect signals in the magnitude of tens of photons at maximum. At that intensity level, the discrete nature of light manifests itself as relatively large fluctuations in the observed photon count (photon shot noise).

In practice, the Poisson distributed photon counting statistics hold for SPAD array sensors if the average photon flux incident on a single pixel is less than one detection event per exposure period [52,53]. For our lidar architecture, this implies that the statistical properties hold up until the signal intensity reaches the limit of 32 photons per channel for a single measurement cycle. Near the saturation limit of the detector, most of the incident photons are absorbed by pixels that have already triggered (the pile-up effect), thus invalidating the assumptions of an ideal detection model. Further, unlike avalanche photodiode (APD)-based linear gain devices, SPAD sensors do not suffer from the avalanche noise, or from the readout noise [50,54]. Therefore, a majority of the noise present in low-photon flux measurements can be accounted for to the Poisson distributed nature of photon arrival times. This has an importance when we derive a model for the spectral reflectance estimation accuracy in the low-photon flux regime.

We begin our derivation by considering a Poisson-distributed random variable with a rate parameter μ. The small sample confidence interval for the Poisson mean can be computed as [55]:(1)Q(α/2,μ,1)≤μ≤Q(1−α/2,μ+1,1)
where Q(b,l,1) denotes the quantile function of the gamma distribution with cumulative probability mass *b*, scale parameter 1, and shape parameter *l*. The variable α denotes the lower and upper tail probability. In addition, the reflectance estimate of the target can be computed as (discussed in [56,57]):(2)$$r_{est}\left(\lambda x\right)=\frac{E\left[S(\lambda x)\right]}{S_{r}\left(\lambda x\right)}\cdot \rho_{r}\left(\lambda x\right)=C\left(\lambda x,d\right)\cdot E\left[S(\lambda x)\right]$$
where $E\left[S(\lambda x)\right]$ is the expected primary target signal intensity for the wavelength channel λx, $S_{r}\left(\lambda x\right)$ denotes the signal intensity of a reference target that is equidistant to the primary target and has reflectance $\rho_{r}\left(\lambda x\right)$. The factor C(λx,d) encapsulates the reflectance calibration values that are dependent on wavelength λx and target distance *d*.

To derive an expression for the confidence limits of channel-wise reflectance values, given a certain observed signal level $E\left[S(\lambda x)\right]$ and a confidence interval (1−α)·100%, we express the upper and lower confidence limits of the expected photon count with the help of the small sample confidence interval given in Equation (Equation 1), and plug the result into the reflectance estimation Equation (Equation 2). Then, the (1−α)·100% confidence interval for the reflectance estimate can be computed as:(3)$$\begin{array}{cc} r_{\begin{array}{c} \mathrm{est}.\mathrm{lower}\\ \mathrm{conf}. \end{array}}\leq \; & r_{est}\leq r_{\begin{array}{c} \mathrm{est}.\mathrm{upper}\\ \mathrm{conf}. \end{array}} \end{array}$$(4)$$\begin{array}{cc} C\left(\lambda x,d\right)\cdot Q\left(\alpha /2,E\left[S(\lambda x)\right],1\right)\leq \; & r_{est}\left(\lambda x\right)\leq C\left(\lambda x,d\right)\cdot Q\left(1-\alpha /2,E\left[S(\lambda x)\right]+1,1\right) \end{array}$$

A more useable form of the above expression can be obtained by computing the ratio of the upper and lower confidence limits to the reflectance estimate (shifted relative error). In the case of overabundant photon counts, the observed reflectance value is overestimated and the upper confidence limit for the ratio is given by:(5)$$\eta_{\mathrm{upper}}=\frac{r_{\begin{array}{c} \mathrm{est}.\mathrm{upper}\\ \mathrm{conf}. \end{array}}}{r_{est}}=\underset{=\;1+\mathrm{relative}\;\mathrm{error}}{\underset{︸}{\frac{Q(1-\alpha /2,E\left[S(\lambda x)\right]+1,1)}{E\left[S(\lambda x)\right]}}}$$

Similarly, when there is a deficit in the photon counts compared to the expected photon flux, the observed reflectance value is underestimated and the lower confidence limit for the ratio is given by:(6)$$\eta_{\mathrm{lower}}=\frac{r_{\begin{array}{c} \mathrm{est}.\mathrm{lower}\\ \mathrm{conf}. \end{array}}}{r_{est}}=\underset{=\;1-\mathrm{relative}\;\mathrm{error}}{\underset{︸}{\frac{Q(\alpha /2,E\left[S(\lambda x)\right],1)}{E\left[S(\lambda x)\right]}}}$$

Figure 4 illustrates the relative upper and lower 95% confidence limits for the reflectance estimate with respect to the number of photon counts. We can observe that, due to the skewness of the Poisson distribution with low rate values, the confidence interval is slightly asymmetric when the photon count is low. Therefore, given an ideal detection model, there is a tendency to rather overestimate the reflectance estimate than to underestimate it in the low-photon count regime. It is also quite evident from the visualization that the photon shot noise dictates the accuracy of the spectral reflectance measurement in single-photon sensitive hyper- and multispectral measurement schemes when the photon flux is small. The results also apply to the reflectance measurement accuracy of monochromatic single-photon sensitive lidars. Despite the fact that the range of the confidence interval of the reflectance estimate is quite broad when the photon counts are small, it is gradually compressed towards zero, the absolute certainty in the estimated reflectance value, as the expected photon count increases.

It is a well-known property of single-photon sensitive imaging schemes that the sample mean of the photon count is close, or equivalent, to the Cramér–Rao lower bound [53]. It can be, thus, argued that the relative error limits given in Equations (Equation 5) and (Equation 6) represent the theoretical reflectance estimation limits for single photon sensitive hyperspectral lidars when the SPAD array is considered to be an ideal detector (100% photon detection efficiency, no dead time, no intrinsic noise, etc.).

### 3.3. Experiments

The objectives of this research are to examine the feasibility of hyperspectral single photon lidar data for autonomous driving purposes and to investigate the limitations of spectral reflectance measurement accuracy that might exist in the low-photon flux regime. Specifically, we compare the statistical model for spectral reflectance measurement accuracy in the low-photon flux regime (introduced in Section 3.2) to sample observations from our prototype lidar, perform a spectral signature separability experiment and investigate the suitability of the data for classification purposes. The two latter experiments aim to determine the extent to which spectral information and very weak return pulse intensity might influence machine learning-based autonomous driving perception algorithms. The dataset used in the experiments has been described in Section 3.3.1.

#### 3.3.1. The Dataset and System Calibration

For the purpose of investigating the feasibility of frame-based single photon sensitive hyperspectral lidar for autonomous-driving-related perception tasks, we collected a dataset consisting of 300 samples in 10 different classes (30 samples from different targets in each class). We use 30 different spectral channels (sensor has 32 channels, but 2 channels were not used), each with a bandwidth of 12.3 nm, from the wavelength band 1200–1570 nm. The dataset classes were selected with the criteria that they should represent some of the most common objects and materials found in the driving environment. The classes include dry asphalt, gravel road, granite, moss, white plaster wall, snow covered asphalt, car body (gray metallic paint), spruce, wet asphalt, and grass. With this dataset we tried to simultaneously explore the properties of the low-photon count regime and the potential that is available with the hyperspectral sensing capability as compared to the commonly available monochromatic lidar technology. To our knowledge, the combination of these two dimensions of measurement are discussed to a smaller extent in previous literature, and are hardly experimented with together at all. To date, several valuable studies [11,33,58] have investigated the distance measurement limitations of single photon sensitive receivers. The topic of distance measurement limitations was deliberately left out of focus in this work when considering the requirements of the dataset, or the experimental setup.

Each sample in the dataset is a set of 10,000 consecutive frames that have been acquired by firing the supercontinuum laser towards the same spatial spot (beam steering was not used) at a pulse repetition rate of approximately 30 kHz (total sample acquisition time was approximately 0.33 s). The spatial location of the target spot, and the incidence angle between the target surface and the laser beam, were widely varied between the sample measurements. The targets were located at distances ranging from 15 to 100 m. The total exposure time was in the order of one μs, while the time-of-flight (ToF) time resolution was set at 250 ps (corresponds to approximately 3.8 cm distance resolution). The dataset was collected in real-life conditions, during the daytime, outside with overcast sky. During the dataset collection, the temperature was approximately two degrees Celsius and the visibility was good (no rain, fog, or snowfall).

In order to perform white balance calibration to the system and to examine the photon counting statistics at the low-photon flux regime, we collected four samples of Spectralon targets in a low ambient irradiance environment. Two Spectralon^®^ diffuse reflectance standard plates were used with reflectance values of 20% and 40%, respectively. The targets were optically flat over the measurement wavelength band with a relative error of ±4% from the nominal reflectance value. The measurement setup was identical to the measurement setup used in the collection of the 10 class dataset; 10,000 consecutive frames were recorded for each sample and the spatial measurement position was static during the acquisition period of the sample. The white balance calibration vector was obtained by first computing the relative spectral reflectance curves of the four Spectralon samples and then taking the average of the resulting spectral curves.

The distance to the Spectralon targets was approximately 16 m in the first set of measurements and approximately 11 m in the second set of measurements. Therefore, in addition to the system dependent white balance calibration values, also the factor originating from the optical properties of the air mass between the lidar and the target plates has been included in the white balance calibration measurements.

In order to explore the magnitude of dark current noise in our measurement data, we collected a set of 100,000 consecutive frames in such a way that any external illumination was prevented from entering the SPAD sensor array (optical path to the sensor was blocked). The magnitude of the dark current noise was approximately $\sigma_{dark}=0.07$ for a time period corresponding to the typical return pulse timer filter window width in our dataset. Because the dark current noise was negligibly small compared to the typical signal photon count, we omitted it completely in the data processing phase.

#### 3.3.2. Spectral Reflectance Measurement Accuracy in the Low-Photon Flux Regime

The theoretical model for the spectral reflectance measurement accuracy in the low-photon flux regime (discussed in Section 3.2) was verified by relying on qualitative visual analysis on the statistical properties of the Spectralon 40% measurements (measurement sample from the white balance calibration measurements). The sequence of 10,000 consecutive frames in the Spectralon sample were split into blocks (a set of $N_{frames}$ consecutive frames) with varying sizes, in the range $N_{frames}\in \left[1,100\right]$, and the photon counts at each block were used to compute a histogram (empirical distribution function) from which the sample estimates for the 95% confidence limits were obtained. The sample estimates for the confidence limits were compared to the theoretical limits, which, in turn, were computed from the expected block-wise photon counts (the expectation value for a single-frame photon count was computed over the full set of 10,000 consecutive frames). As the Spectralon sample measurements were carried out in low ambient light environment, we expect that the empirical results should adhere closely to the theoretical model. This is because the majority of detected photons should originate from the coherent illumination pulse of the laser source, instead of an external light source with possibly super-Poissonian photon counting statistics.

#### 3.3.3. Separability of the Hyperspectral Single Photon Data

Current autonomous driving perception systems rely heavily on machine learning methods, such as semantic segmentation and bounding box object recognition, where the objective is to predict a set of class probabilities from the input data [59]. The performance of these algorithms is inherently limited by the separability of the input data either in the original measurement space, or in some higher or lower dimensional space into which the data have been transformed into.

In order to explore the degree of separability in our data, we used the t-distributed stochastic neighbour embedding (t-SNE) [60] algorithm to project the data into a two-dimensional embedded space for visual verification. The output embedding of the t-SNE algorithm provides a fairly reliable estimate of the inter-class differences based on the sample features, which arguably also indicates the separability of the data to a certain degree. However, due to the nature of the algorithm, the intra-class cluster shapes and the absolute point distances in the embedded space should be interpreted with caution and should not be used to draw any conclusions about the underlying structure of the data generating distribution [61].

As an input data X to the t-SNE algorithm, we used the white balance normalized relative reflectance spectra (area under the curve normalized to unity) from all 300 samples (10 classes, 30 samples in each class):(7)$$X={\left[S_{\mathrm{wbn}}^{\left(1,1\right)}\;,\;S_{\mathrm{wbn}}^{\left(1,2\right)}\;,\;\ldots \;,\;S_{\mathrm{wbn}}^{\left(10,30\right)}\right]}^{⊺}$$
where $S_{\mathrm{wbn}}^{\left(c,i\right)}$ corresponds to the white balance normalized relative reflectance spectrum of the *i*’th sample in class *c*. The relative reflectance spectra was used instead of the regular reflectance spectra in order to concentrate the analysis on the spectral shape dependent properties of the data without adding an external error source due to calibration related non-idealities. The data were normalized before feeding into the t-SNE algorithm by mean-centering and rescaling the spectral channels to unit variance:(8)$$Z_{n}\left(\lambda x\right)=\frac{X_{n}\left(\lambda x\right)-\bar{x}\left(\lambda x\right)}{\bar{\sigma }\left(\lambda x\right)}$$
where $\bar{x}$ and $\bar{\sigma }$ denote the column-wise mean- and standard-deviation vectors of X, respectively, and *n* denotes the row index of X.

We computed the t-SNE embedding for three different block sizes ($N_{frames}\in \left\{1,10,200\right\}$) in order to capture the effect of the spectral reflectance measurement accuracy on the separability of the data. Full spectral resolution was used ($N_{channels}=30$) at each of the three tests. We used the t-SNE implementation from the scikit-learn library [62] with the learning rate set at lr=10.0, the number of iterations at $N_{iter.}$ = 50,000 and the perplexity at perplexity=5.0.

#### 3.3.4. Classification with Random Forest Classifier

We ran a classification experiment where a random forest classifier [63] was trained to classify the dataset samples in their respective classes while the photon count (block size) and the spectral resolution (number of binned channels) was varied. The main idea behind the experiment was to verify the degree to which machine learning methods are able to extract useful information from the spectral measurements at the challenging low-photon count regime, where the feature vectors are noisy due to photon shot noise. Additionally, we found it important to test the impact of spectral resolution on the classification accuracy (this was implemented by varying the number of binned channels, while the spectral band remained the same), because it is likely that the information content of the spectral dimension has further positive implications for the robustness of more sophisticated high-capacity machine learning methods, such as convolutional neural networks (CNNs).

We used the white balance normalized relative reflectance spectra (area under the curve normalized to unity) of each class in the autonomous-driving-related dataset (see Section 3.3.1) as an input data to the classifier. Similarly to the separability experiment, the relative reflectance spectra were selected in order to reduce the impact of non-idealities from the reflectance calibration process and to concentrate the analysis on the spectral-shape-dependent properties of the data instead. Adding the missing degree of freedom from the absolute reflectance values can be expected to improve the results moderately.

Due to the small size of the dataset, 5-fold cross-validation was applied. At each fold, the training set size was 240 samples (24 samples per class) and the test set size 60 samples (6 samples per class). Two hyperparameters were varied between the tests: the block size (the photon count was accumulated over a block of multiple consecutive frames) was altered in the range $N_{frames}\in \left[1,\;10,000\right]$ in order to examine the influence of the spectral reflectance measurement accuracy, and the number of spectral channels was varied according to $N_{channels}\in \left\{30,15,10,6,5,4,3,2\right\}$ by applying channel binning. The binned wavelength bands were equally wide when channel binning was applied, except when the number of channels was set at $N_{channels}=4$. In this situation, we split the original channels into new channels by using slightly wider bin width of $N_{bins}=8$ on the two central channels, while the bin width was set at $N_{bins}=7$ on the border channels.

The random forest classifier was trained from the ground up for each spectral resolution value (number of channels), and for each cross-validation fold. In the training phase, the training samples consisted of the white balance normalized relative reflectance spectra that had been accumulated over the full range of 10,000 consecutive frames per sample. This was completed in order to ensure that the classifier would learn, for each class, an estimate of the spectral signature that would be as close as possible to the underlying noise-free spectral signature.

The experiments were carried out by using the random forest classifier implementation from the scikit-learn library [62]. The number of trees in the random forest classifier was set at 100, based on the suggestion in [64], and no further tuning of the model parameters was carried out. The performance of the random forest classifier instances was evaluated by using the accuracy metric:(9)$$\mathrm{Accuracy}\left(y,\hat{y}\right)=\frac{1}{N_{fold}}\sum_{k=1}^{N_{fold}}\left(\frac{1}{N_{samples}}\sum_{i=1}^{N_{samples}}⟦y_{k}^{\left(i\right)}={\hat{y}}_{k}^{\left(i\right)}⟧\right)$$
where $y_{k}^{\left(i\right)}$ and ${\hat{y}}_{k}^{\left(i\right)}$ denote the ground truth class and the predicted class at fold *k* and with the test set sample index of *i*, respectively. $N_{fold}$ denotes the number of cross-validation folds and $N_{samples}$ denotes the number of test set samples. We do not resort to more complex performance metrics, because the class sizes in the dataset are in good balance, which enables the accuracy metric to provide sufficient insight into the performance of the trained models.

The accuracy of the trained random forest classifiers is compared to a baseline accuracy of 10% which can be obtained by predicting the sample class by randomly selecting it from a discrete uniform distribution over the class labels.

### 3.4. Data Processing

The following sections describe the process of estimating the target distance, computing the relative spectral reflectance curve from a single frame, or from a sequence of measurements, and reducing the spectral resolution by channel binning. The data processing pipeline has been visualized in Figure 5.

#### 3.4.1. Spectrum Measurement from a Single Frame

Each measurement cycle with our hyperspectral single photon lidar produces a data frame I that contains 32 × 32 delay counter values (in this study, only 30 spectral channels are used among the 32 channels available) representing the time-of-flight time differences $\Delta t=t_{1}-t_{0}$ between the supercontinuum laser triggering time at $t_{0}$ and the return pulse detection time at $t_{1}$ as:(10)$$I_{ij}^{\left(k\right)}=\Delta t_{ij}$$
where *i* and *j* denote the pixel indices in the intensity and wavelength directions, respectively, and *k* denotes the location of the frame in a sequence of measurements. The pixels are triggered either by photons originating from the target return pulse or ambient illumination source, or alternatively by SPAD array intrinsic noise, such as dark current noise [54], afterpulsing [65], or crosstalk [66,67,68,69]. The pixels in the SPAD array that have not triggered during the measurement cycle are denoted by value Δt=0.

The target distance was estimated by a following process: first, a timer histogram was computed from the SPAD array pixel counter values I, and then a Gaussian single-target reflection model g(x) (given in Equation (Equation 11)) was fitted to the histogram by minimizing the sum of least squares with the Levenberg–Marquardt algorithm. We have assumed that the return waveforms are dominated by the echo from the primary target and use the most prominent histogram peak as an initial guess in the Gaussian single-target reflection model:(11)$$g\left(x\right)=const.+A\cdot \exp\left(-\frac{{\left(x-\mu_{dist.}\right)}^{2}}{2\sigma_{fw}^{2}}\right)$$
where *x* denotes the timer histogram values, $\mu_{dist.}$ denotes the time-of-flight distance in camera clock cycles, $\sigma_{fw}$ denotes the width of the return pulse and *A* is the signal intensity dependent amplitude scaling factor. The constant term const. represents the ambient illumination flux that is assumed to be fairly uniform over the signal acquisition period.

Once the estimates for the distance $\mu_{dist.}$ and the pulse width $\sigma_{fw}$ were computed, temporal filtering was applied and the photon count for each spectral channel λx was evaluated as:(12)$$S_{\lambda x}=\sum_{i=1}^{32}⟦⟦I_{i,\lambda x}>0⟧\wedge ⟦I_{i,\lambda x}<\left(\mu_{dist.}+3\sigma_{fw}\right)⟧\wedge ⟦I_{i,\lambda x}>\left(\mu_{dist.}-3\sigma_{fw}\right)⟧⟧$$
where the wavelength of channel λx is measured at the center of the SPAD array pixel row corresponding to that channel. In Equation (Equation 12), we have used the Iverson bracket notation:(13)$$⟦K⟧=\left\{\begin{array}{cc} 1 & \mathrm{if}\;K\;\mathrm{is}\;\mathrm{true}.\\ 0 & \mathrm{if}\;\mathrm{otherwise}. \end{array}\right.$$

When operating in a low-photon flux environment, it is important to capture most of the signal photons due to the inherently low signal-to-noise ratios. Therefore, the timer value filter window $\mu_{dist.}\pm 3\sigma_{fw}$ was deliberately chosen to be relatively wide, in order to take into account the broadening of the supercontinuum laser illumination pulse due to chromatic group velocity dispersion [47,70,71] and to capture the pulse broadening effect of the target impulse response function.

Following the computation of the channel-wise photon count $S_{\lambda x}$, a vector S was constructed that is an discrete approximation of the spectrally dispersed return pulse photon flux density incident on the SPAD array:(14)$$S=\left[S_{\lambda 1}\;,\;\ldots \;,S_{\lambda 30}\right]$$

In order to obtain an estimate of the shape of the spectral reflectance curve $r_{est}$ at the measured wavelength band without resorting to extensive calibration measurements, and to obtain feature vectors that are comparable at different target distances, the photon count in each channel was first normalized by the total photon count over all channels:(15)$$S_{n}\left(\lambda x\right)=\frac{S(\lambda x)}{\int_{\lambda 1}^{\lambda 30}S\left(\lambda y\right)\;d\lambda y}$$

Then, the normalized photon count vector $S_{n}$ was divided element-wise by the white balance calibration vector $S_{n}^{\left(wb\right)}$ (which was obtained from the Spectralon measurements and was also normalized according to Equation (Equation 15)) in order to obtain the relative spectral reflectance curve:(16)$${\tilde{r}}_{est}=S_{n}\cdot diag^{-1}\left(S_{n}^{\left(wb\right)}\right)=\left[S_{n,\lambda 1},\ldots ,S_{n,\lambda 30}\right]\left[\begin{array}{ccc} \frac{1}{S_{n,\lambda 1}^{\left(wb\right)}} & \; & \\ \; & \ddots & \\ \; & \; & \frac{1}{S_{n,\lambda 30}^{\left(wb\right)}} \end{array}\right]$$

The spectral reflectance curve obtained in this way is relative, in the sense that the scale does not represent the true target reflectance value, but the shape of the curve resembles the shape of the absolute spectral reflectance curve $r_{est}$.

In expression (Equation 15), the denominator is also subject to Poisson-distributed photon shot noise, which influences the upper and lower confidence intervals for the spectral reflectance measurement accuracy discussed in Section 3.2. If we accept a small approximation error at the low-photon count regime, we can omit the fluctuations in the denominator given that the total photon count over the spectral channels is large enough, in the order of tens of photons or more (on our device, the total photon count is expected to be at minimum 30, if at least one photon per channel has been detected).

#### 3.4.2. Signal Acquisition over Consecutive Frames

The maximum signal capacity for a single frame measurement on our hyperspectral single photon lidar has been limited by the SPAD array resolution to 32 photons per channel. The limitation was addressed in the experiments by artificially increasing the signal photon count by employing block-wise signal accumulation scheme where the photon count from a block of two or more consecutive frames was combined together:(17)$$S_{\mathrm{block}}^{\left(i\right)}\left(\lambda x\right)=\sum_{k=i}^{i+N_{frames}}S^{\left(k\right)}\left(\lambda x\right)$$
where indices *i* and *k* represent the location of the frame in the measurement sequence, and $N_{frames}$ denotes the number of consecutive frames.

In the block-wise signal accumulation scheme, the expected photon count scales linearly with respect to the number of frames:(18)$$E\left[{S_{\mathrm{block}}}^{\left(i\right)}\left(\lambda x\right)\right]=N_{frames}\cdot E\left[S(\lambda x)\right]$$

This allows the signal-to-noise ratio (SNR) in the block-wise signal accumulation scheme to be expressed as:(19)$${\mathrm{SNR}}_{\begin{array}{c} \mathrm{est}\\ \mathrm{block} \end{array}}\left(\lambda x\right)=\frac{N_{frames}\cdot E\left[S(\lambda x)\right]}{\sqrt{N_{frames}\cdot E\left[S(\lambda x)\right]}}=\sqrt{N_{frames}}\cdot {\mathrm{SNR}}_{est}\left(\lambda x\right)$$
where ${\mathrm{SNR}}_{est}\left(\lambda x\right)$ refers to the SNR estimate of a single frame measurement at channel λx. The block-wise SNR scales with respect to the square root of the block-size $N_{frames}$, which implies that the experimental results might be the most sensitive to variations in the signal photon count when the block-size $N_{frames}$ is small (magnitude of the SNR gradient is the highest).

#### 3.4.3. Channel Binning

Channel binning was applied in the classification experiment in order to capture the effect of the spectral resolution in the classification accuracy. The channel binning operation was carried out by summing the photon counts at two or more adjacent channels together:(20)$$\left\{\begin{array}{c} S_{\mathrm{binned}}\left(\lambda y\right)=\sum_{\lambda x\in \Lambda }S\left(\lambda x\right)\\ \Lambda =\{\lambda k\;,\;\lambda \left(k+1\right)\;,\;\ldots \;,\;\lambda \left(k+N_{bins}-1\right)\}\\ \lambda y=\sum_{\lambda x\in \Lambda }\frac{\lambda x}{\left|\Lambda \right|} \end{array}\right.$$
where λy refers to the central wavelength of the new binned spectral channel, $N_{bins}$ denotes the bin width and the index *k* refers to the left-edge of the binned channel in the full spectral resolution coordinates. The index *k* was selected, such that the channel binning operation was applied only to non-overlapping channels in the original spectrum S.

The effect of channel binning on the spectral resolution has been illustrated in Figure 6. It is quite evident that increasing the bin width of the binned spectral channels (lower number of channels in total) reduces the capability of the binned spectral curve to approximate the underlying spectral signature of the target, but at the same time increases the amount of signal per channel.

## 4. Results

### 4.1. The Dataset and Calibration Measurements

The autonomous-driving-related dataset has been visualized in Figure 7. The visualization shows examples of the dataset classes along with their respective relative spectral reflectance curves that have been averaged over $N_{frames}$ = 10,000 consecutive frames. The intra-class sample spectra show a high degree of similarity, and for most of the dataset classes, the spectra have been closely bunched together with minor variations between the samples. On the contrary, each dataset class has an easily characterizable spectral curve with distinctive shape compared to the other classes. There are exceptions however, for example, the spectral curves between the classes “gravel road” and “dry asphalt” have remarkably similar shapes and are almost indistinguishable from each other.

Figure 8 illustrates the characteristic relative spectral reflectance curves that have been calculated as an average over the 30 samples for each respective dataset class. Additionally, the effect of the photon shot noise on the spectral curves has been demonstrated by computing the spectra for four different block sizes. In the single frame measurement ($N_{frames}=1$), the photon shot noise causes significant variations in the spectra, but the general shape of the spectra can still be observed. As the block size increases, the spectra begins to slowly resemble the underlying noise-free spectral signature.

Figure 9 illustrates both the class-wise average photon count for the whole exposure period, and the class-wise average photon count for the target return pulse. The target photon count for all dataset classes resides comfortably in the low-photon flux regime, ranging from approximately 1 to 10 photons per channel for a single frame. The photon count over the whole exposure period does not show signs of sensor saturation.

The visualization in Figure 10 illustrates the return waveforms from the spruce target at different wavelength channels. In addition to the dominant return pulse (at approximately 330 ns), the return waveforms also capture the trees that are obstructed by the primary target. The return echoes at different wavelength channels appear to have high temporal correlation, although the echo shapes are not exactly identical across the spectral range.

The relative spectral reflectance curves of the Spectralon measurements have been visualized in Figure 11. Similarly to the dataset spectra, the Spectralon spectra has also been substantially affected by the photon shot noise at the low-photon flux regime. When the relative reflectance spectra have been computed over $N_{frames}$ = 10,000 consecutive frames, the spectral curves have converged quite close to an ideal (“flat”) white balance spectrum.

In order to investigate the convergence of the Spectralon spectra towards the ideal white balance spectrum, the root-mean-square error (RMSE) was calculated between the Spectralon spectra and the ideal white balance spectrum with respect to the block size. The resulting error graph can be observed in Figure 12. It can be seen that the RMSE starts to plateau already when the number of consecutive frames has reached $N_{frames}$ = 1000. Therefore, it can be expected that increasing the frame count past 10,000 consecutive frames does not substantially increase the accuracy of the white balance calibration vector.

The average block-wise photon count of the Spectralon 40% sample, in addition to the sample standard deviation of the block-wise photon count, has been visualized in Figure 13. The results show quite evidently that the shapes of the sample standard deviation curves do not perfectly correlate with the shapes of the photon count curves (there should be a quadratic correspondence in the magnitude), which would be the case, if the ideal Poisson distributed assumption for the photon counts would hold. The magnitudes of the sample standard deviation values lead us to believe that the underlying signal photon count, especially in the wavelength range from 1200 nm to 1350 nm, is in reality slightly lower than the photon count values at the left-hand side figure show. This observation implies that there is small amount of hardware-related systematic bias in the photon counts.

Additionally, the water vapour absorption peak in the air mass can be observed in Figure 13 as a local minimum in the photon counts at the wavelength band from 1400 nm to 1450 nm.

### 4.2. Spectral Reflectance Measurement Accuracy in the Low-Photon Flux Regime

Figure 14 presents the relative spectral reflectance curves of the Spectralon 40% sample along with the theoretical 95% confidence interval for various block sizes. The observations are in accordance with the theory: a majority of the sample spectra lie comfortably inside the confidence interval limits and only a few observations exceed the confidence interval limits at some wavelength bands. Additionally, it can be observed that most of the probability mass has been concentrated towards the mean of the spectrum leaving a fair margin to the upper confidence limit.

The theoretical relative confidence limits $\eta_{\mathrm{lower}}$ and $\eta_{\mathrm{upper}}$ along with the sample estimates have been illustrated in Figure 15. The sample estimates have been computed from the empirical distribution function for block-wise photon count. The results have been visualized for the wavelength channel λx=1557 nm, that in our measurement data is closest to the 1550 nm wavelength band (common operating wavelength of InGaAs—based sensor), but the results are identical throughout the spectral range. Additionally, we have visualized the channel-wise signal-to-noise ratio (SNR) as a function of the block size.

The visualization reveals that the sample observations for the lower confidence limit seem to be in agreement between the theoretical values. However, the theoretical upper confidence limit has been estimated slightly more conservatively when compared to the confidence limit given by the sample observations. A similar tendency to overestimate the theoretical upper confidence limit when compared to the sample observations was also observed in Figure 14.

The visualization in Figure 15 reveals, also, the fallibility of the reflectance measurement accuracy in conditions where the SNR would be high enough to resolve the target distance with a fairly high reliability. For instance, a single frame measurement, given the conditions of the visualization, provide an approximate signal-to-noise ratio of SNR≈3 at the wavelength channel λx=1557 nm, which is more than enough to estimate the target distance with relatively high accuracy. However, at the same time the relative reflectance measurement error is close to ±70% (sample observations), which reduces the informativeness of the estimated reflectance value significantly.

### 4.3. Separability of the Hyperspectral Single Photon Data

The results of the dataset separability experiment have been illustrated in Figure 16. In the case of a single frame measurement, the dataset classes mostly reside in a single cluster, but it is possible to observe samples belonging to certain classes separating far away from each other. For instance, the classes “white wall” and “snowy asphalt” are located in the opposite quadrants of the embedded space, which enables them to be separated linearly. As the block-size is increased to 10 frames, the spectral reflectance measurement accuracy has improved to a point where the samples start to separate into their own clusters, although there is still a substantial amount of intermixing between the classes. Finally, when the spectra have been calculated over 200 consecutive frames, the dataset classes have mostly separated into their class specific clusters. An exception to this are the classes “wet asphalt” and “grass”, which form clusters that are very close to each other and are partly mixed. Similar behaviour can be observed for the classes “gravel road” and “dry asphalt”, and also, for the classes “granite” and “white wall”.

### 4.4. Classification with Random Forest Classifier

The purpose of the classification experiment was to demonstrate the use of the hyperspectral single photon lidar data in an autonomous-driving-related perception task, and, also, to establish the feasibility of the data for classification purposes in the low-photon flux regime. The results of the experiment have been visualized in Figure 17.

The mean classification accuracy in the test set reflects the theory in multiple ways: First, the rate of improvement in the classification accuracy is the highest when the block size is relatively small, while the rate of improvement stagnates when we move towards larger block sizes. The behaviour reflects closely the asymptotically convergent shape of the relative reflectance confidence limits. In the low-photon flux regime the confidence limits substantially improve as the signal photon count is increased, and then, at the high-photon flux regime, shrink towards better reflectance estimation certainty in much smaller increments. Second, at the high-photon flux regime ($N_{frames}\geq 10^{2}$), the classification accuracy is mostly dictated by the number of spectral channels. Applying channel binning provides a slight advantage to the classification accuracy with small block sizes by increasing the channel-wise photon count, but the advantage is lost as the signal levels increase to a point where the photon shot noise has only a small overall contribution to the feature vector variability.

It should be pointed out that it is possible to achieve mean classification accuracies between 30% and 38%, depending on the number of channels, even with a single frame measurement, which certainly can be considered to be in the low-photon flux regime. When compared to the baseline model with an accuracy of 10%, the improvement is significant. However, the classification accuracies can be considered good (accuracies of over 70%) only when the block size is extended above 10 frames, which translates into signal photon counts in the order of $10^{2}$ photons per wavelength channel.

## 5. Discussion

### 5.1. Spectral Reflectance Measurement Accuracy in the Low-Photon Flux Regime

One of the initial objectives in this study was to find theoretical limits to the spectral reflectance measurement accuracy in the low-photon flux regime. The simple theoretical model that was derived based on the quantile function of the gamma distribution fits the data relatively well, although the higher confidence limit was overly conservative when compared to the sample observations with small number of photon counts. The difference of the derived model to the measurement results might be explained by the pile-up effect [72], which distorts the SPAD array intensity data by under-estimating the true photon count. Therefore, the upper confidence limit that was calculated from the empirical distribution function has been estimated at a slightly lower level than the theory leads us to believe.

The pile-up effect exists only for high-intensity return pulses, which explains why the lower confidence limits are in agreement between the theory and the sample observations. Further, the expected frame-wise photon count $E\left[S\right]$ can be considered to be less affected by the pile-up effect than the upper percentiles of the empirical distribution function, because the latter has a higher signal level, which increases the probability of pile-up. This would imply that the confidence limits provided by the theoretical model, and that were calculated using the expected frame-wise photon count $E\left[S\right]$, are closer to the underlying true spectral reflectance measurement accuracy limits.

For the purpose of computing the spectral reflectance curve as accurately as possible from the data, it is recommended to use a proper photon counting statistics model [65,73] that incorporates the intrinsic SPAD array noise and bias sources, such as afterpulsing, crosstalk, and the pile-up effect. The fluctuating level of ambient photon counts in the signal intensity should also be taken into account in a more detailed analysis. Our statistical model for the spectral reflectance measurement accuracy can be used as a worst case upper bound for the relative error of the reflectance estimate when the measurement conditions are fairly close to the ideal detection model.

### 5.2. Hyperspectral Single Photon Data Separability and Feasibility for Classification Purposes

The results of the dataset separability experiment indicate that the main factor contributing to the separability of the hyperspectral single photon data is, most probably, the spectral reflectance measurement accuracy, which, in turn, is highly dependent on the return pulse photon count. In many cases, the spectral signature of important targets differs substantially across the target classes [74]. It can, therefore, be assumed that the limiting factor in the data separability is, indeed, the accuracy at which the spectral signatures can be measured. In order to achieve a perception pipeline that is both accurate and robust, also in the low-photon flux regime, the fundamental limitation due to photon shot noise has to be taken into account.

It was observed in the experiments that a few of the dataset classes were intermixed in the t-SNE embedding even in a scenario where the spectral reflectance measurement accuracy was adequately high. This is quite natural when the underlying material properties in the measurement setup are considered more closely. For example, at the time of the dataset collection the grass field was slightly moist, which, to a certain degree, explains the similarity of the spectral features of the “grass” class when compared to the “wet asphalt” class. Likewise, the dry asphalt surface that was used as one of the dataset targets was deteriorated due to age, which made the gravel particles in the surface course stick out. The gravel particles provided a reflective surface area for most of the illumination pulse photons instead of the asphalt matrix material, which, in turn, made the spectral signature of the “dry asphalt” class resemble the “gravel road” class.

Further study is required to investigate the degree to which the noise properties of single point measurements affect high-capacity machine learning methods, such as deep learning networks, when the input data are a point cloud instead of a set of individual points. It might be possible that the internal representations formed by the deep learning networks learn to perform, in a sense, signal accumulation over finite spatial regions with object borders declared by the geometry of the object. The internal representations could in principle, with high enough spatial point density, achieve a similar noise reducing effect that was achieved with the block-wise measurement scheme, but without requiring the measurement cycle to be run multiple times for a single spatial location.

The impact of the spectral reflectance measurement accuracy on the dataset separability could also be observed in the results of the classification experiment. Although the dataset was small, the strongly increasing trend in the classification accuracy with respect to the block size (correlates with the channel-wise photon count) was still clearly visible. Additionally, it is interesting to note that the results of the classification experiment support the idea that high spectral resolution is preferable to low spectral resolution, even when the reduced spectral resolution would account to higher spectral reflectance measurement accuracy at the binned channels, at least at the wavelength band used by our lidar. Channel binning is therefore not advisable without optimizing the individual channel bin widths in such a way that the spectral signature separability is maximized simultaneously to minimizing the channel-wise photon shot noise. This process is naturally heavily dependent on the application, and is therefore an important subject for further study.

### 5.3. Principal Implications for Autonomous Vehicle Perception Systems

The findings in this study, while preliminary, may help us to better understand the challenges and strengths associated with next-generation automotive lidar technology. In earlier research it has been noticed that many autonomous-driving-related perception methods benefit from the use of lidar reflectance or intensity channel information in conjunction to the 3D point cloud coordinate information, even when the lidar operates on a monochromatic basis [75,76,77,78]. The research has been conducted by employing avalanche photodiode (APD) based lidar technology, for which the return pulse photon counts are at least two orders of magnitude higher compared to single photon sensitive lidar devices, and, therefore, the intensity channel data have not been affected considerably by photon shot noise. Single photon technology has already been applied at the Ouster OS automotive and robotics lidar sensor family [11,79] and the apparent benefits of the technology [7,80] might further increase the rate of adoption in commercial, mass-produced, devices. Therefore, it is important to consider the limitations to the spectral reflectance measurement accuracy in the low-photon flux regime, even for single-wavelength single photon sensitive lidars.

The advantages of the hyperspectral single photon lidar architecture introduced in this study include the low-complexity of the data processing phase and the ability to simultaneously measure the spectral signature over the whole wavelength band from a single spatial location with only a single illumination pulse. In this regard, the lidar architecture, when combined to a beam steering unit, would be suitable for use in autonomous vehicles, where achieving minimal latency from the target detection to the control system output is often desired. There are, however, some disadvantages that might reduce the practical operating envelope of the sensor, such as the loss of SPAD array capacity in long-range measurements due to the accumulation of dark noise- or ambient photon triggered pixels, and the limited dynamic range with current, fairly low resolution, pixel-wise TDC-based SPAD arrays.

The issue of reduced SPAD array capacity at long-range measurements could be addressed either by delaying the initialization of the sensors TDC counters after a trigger pulse has been detected, or by operating the lidar in range-gated mode [81,82,83]. Current, state-of-the-art, range-gate enabled SPAD arrays [49,50] have much higher resolution than the sensor used in this study, which would also increase the available dynamic range of the measurement substantially. In addition, the range-gated operation would enable the imaging through semi-transparent surfaces [50]. This feature would be especially helpful in urban environments, where different types of transparent building elements are ubiquitous. In order to maintain the ability to measure the spectral signature rapidly without excessively extending the acquisition period for a single point measurement, the distance to the primary target could be resolved in advance by using a probing pulse, and only then perform a range-gated spectral measurement from the probed depth with a small number of range-gates, or possibly with only a single range-gate.

The high-sensitivity of single photon detectors provides the possibility to combine functionalities of camera and lidar devices in the same sensor. This has been already implemented in multiple Ouster lidar models [79], which provide the *ambient* channel for each point cloud point. With spectrograph based hyperspectral single photon lidar, the lidar device could also be used as a hyperspectral camera. This would be beneficial in situations, where, for example, due to high solar irradiance, the normal operating conditions of the lidar device would have been deteriorated.

## 6. Conclusions

In this feasibility study, a frame-based single photon sensitive hyperspectral lidar has been developed and introduced in the context of autonomous vehicle perception. The lidar architecture allows the simultaneous measurement of geometric and spectral information from each point with a single supercontinuum laser illumination pulse. The measurement approach might be less prone to motion blur, require less illumination power on average, and also be computationally less demanding than comparable actively illuminated spectral measurement methods.

We introduce a statistical model for estimating the accuracy of the spectral curve in the low-photon flux regime, and demonstrate how the model can be used to calculate confidence bounds of the relative spectral reflectance curves of single photon sensitive spectral reflectance measurements. Further, we explore the separability of the single photon hyperspectral lidar data as a function of the photon count with a focus on increasing the robustness of autonomous vehicle perception systems. Finally, we test the feasibility of the data for classification purposes in an autonomous-driving-related classification task.

The results show that the statistical spectral reflectance accuracy model conforms closely to the observations, disregarding the SPAD sensor non-idealities. The model indicates that in order to obtain adequately high spectral reflectance measurement accuracy, with relative error of less than ±10% at significance level of α=0.05, the photon count must be at minimum in the order of $10^{2}$ photons per wavelength channel, which is still less than half of the detection threshold (approximately 250 photons) of a survey-grade linear-mode lidar [84].

The separation of the data into class-specific clusters in the 2D t-SNE embedding was observed to be highly dependent on the amount of photon shot noise in the spectral curves. A few of the dataset classes were already separated into class specific clusters when the signal levels were in the order of 10 photons per wavelength channel, but the unambiguous delineation of most of the class clusters was achieved only after the signal level reached approximately $10^{2}$ photons per wavelength channel.

The results of the classification experiments imply that the measurement data provide useful information about the material specific spectral signatures, even with relatively low photon counts of less than 10 photons per wavelength channel. For a single frame measurement (maximum intensity of 32 photons), the mean classification accuracy in our dataset with a random forest classifier was approximately 35% with 30 wavelength channels, which is a substantial improvement over the baseline model with an accuracy of 10%. However, achieving mean classification accuracy of over 90% with the full spectral resolution requires the detection of multiple hundreds of photons per wavelength channel, which can be accomplished by using high laser repetition rate and block-wise measurement scheme, or by increasing the SPAD array resolution.

Additionally, it was found that increasing the spectral resolution improves the classification accuracy, not only in the high-photon count regime, but also when the photon count is low. Therefore, we believe that hyper- and multispectral lidar devices, single photon sensitive or not, can increase the robustness and accuracy of many lidar based autonomous driving and robotics perception methods in the future.

## Figures and Tables

**Figure 1 sensors-22-05759-f001:**
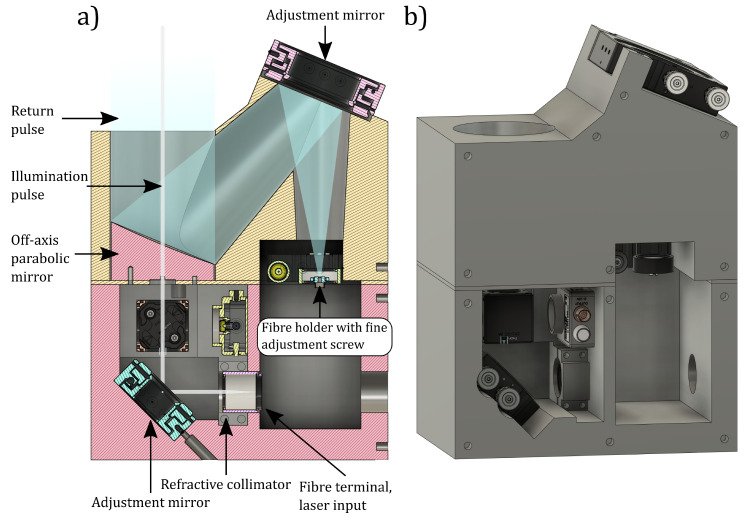
The optical head assembly (**a**) cross-section view and (**b**) 3D model.

**Figure 2 sensors-22-05759-f002:**
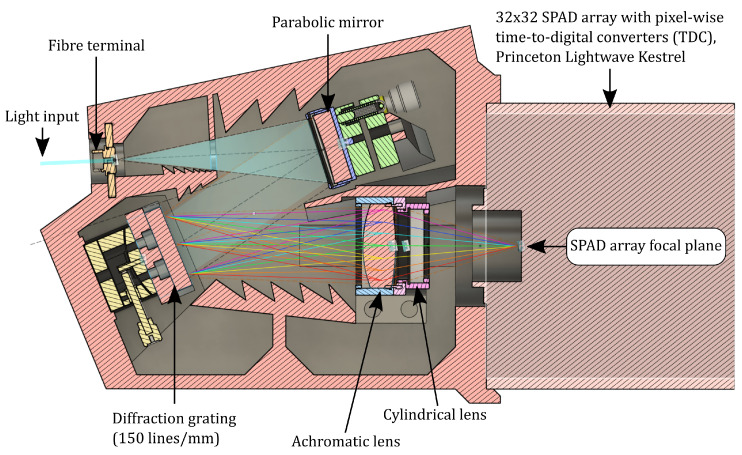
Cross-sectional view of the spectrograph assembly. The spectrally separated return pulse photons are passed through a cylindrical lens to distribute the wavelength bands along a single axis on the SPAD array, while the intensity information is recorded on the second axis.

**Figure 3 sensors-22-05759-f003:**
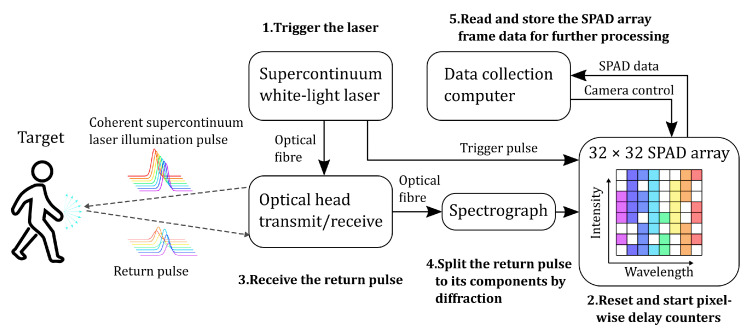
Operational block diagram of the hyperspectral single photon lidar.

**Figure 4 sensors-22-05759-f004:**
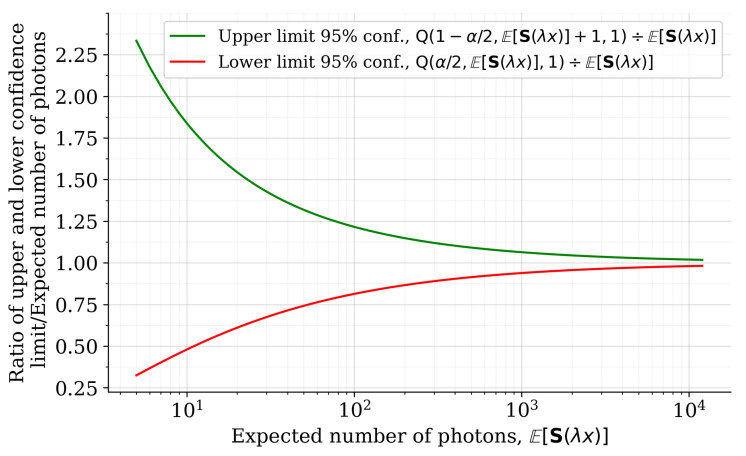
The relative 95% confidence interval (α=0.05) of the reflectance estimate with respect to the expected number of photons per channel. Both the upper limit $\eta_{\mathrm{upper}}$ (green curve) and the lower limit $\eta_{\mathrm{lower}}$ (red curve) converge towards unity (absolute reflectance estimation certainty) as the photon count increases.

**Figure 5 sensors-22-05759-f005:**
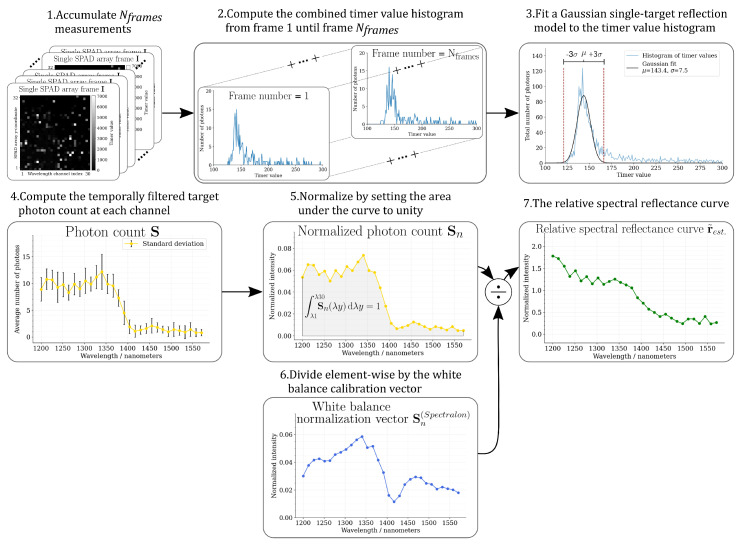
Step-by-step process for computing the relative spectral reflectance curve ${\tilde{r}}_{est}$. The photon counts S are normalized by setting the area under the curve to unity ∫S(λy)dλy=1. Due to the normalization approach, the shape of the relative spectral reflectance curve ${\tilde{r}}_{est}$ estimates the material specific spectral reflectance curve $r_{est}$ without requiring the use of an additional calibration step for calculating the absolute reflectance values.

**Figure 6 sensors-22-05759-f006:**
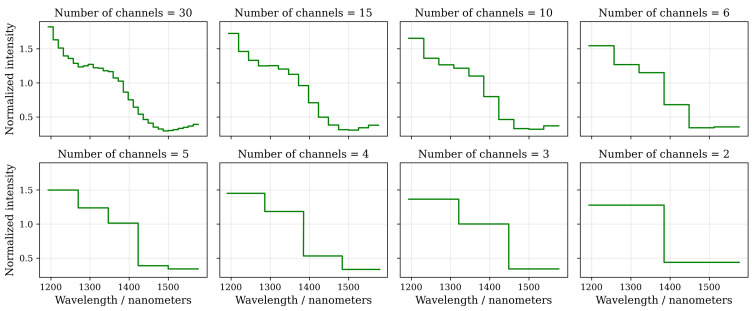
An example of channel binning with various bin widths $N_{bins}$ (sample class grass, normalized spectrum over 10,000 frames). The ability of the binned spectrum to approximate the original spectrum suffers as the bin width is increased (number of channels is reduced). On the other hand, the channel-wise signal level increases, improving the signal-to-noise ratio.

**Figure 7 sensors-22-05759-f007:**
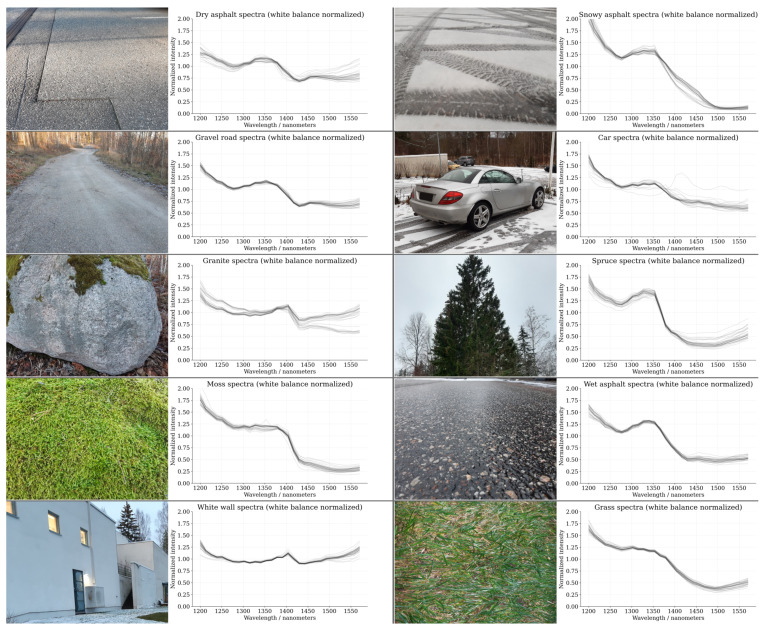
Examples of the dataset classes and their respective relative spectral reflectance curves (each curve represents one of the class specific measurements from a total of 30 per class). The spectral curves have been averaged over the full measurement sequence of 10,000 frames ($N_{frames}$ = 10,000). The dataset consists of 10 classes with 30 samples in each class (300 samples in total). Each sample has been acquired as a static spot measurement (no beam steering) by recording 10,000 consecutive frames (≈1 μs exposure time per frame) from the SPAD array (0.33 s acquisition time per sample at 30 kHz laser pulse repetition rate).

**Figure 8 sensors-22-05759-f008:**
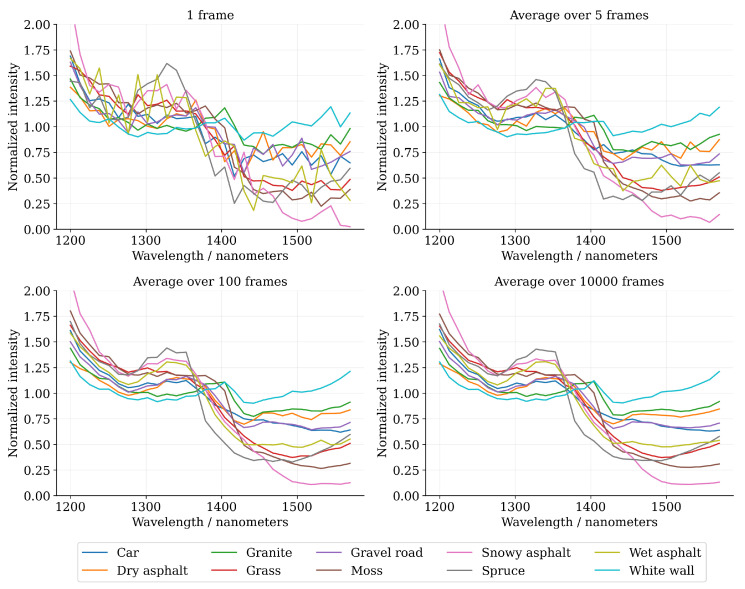
Visualization of the characteristic (average over all 30 samples in each class) relative reflectance spectra with respect to block size. The spectra have been computed over a block of consecutive frames with block sizes $N_{frames}\in \left\{1,5,100,10,000\right\}$. The spectral curves are noisy due to photon shot noise in the low-photon flux regime, but the noise gradually reduces as the block-size increases.

**Figure 9 sensors-22-05759-f009:**
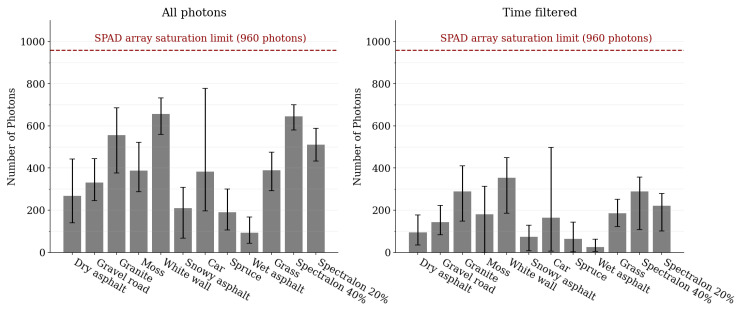
The average photon count for each sample class (total photon count over all wavelength channels). The left-hand side shows the photon count during the whole exposure period while the right-hand side shows the photon count for the target return pulse (number of detections where the timer values are in the range $\left[\mu_{dist.}-3\sigma_{fw}\;,\;\mu_{dist.}+3\sigma_{fw}\right]$). The error bars denote the maximum and minimum photon count within the sample class.

**Figure 10 sensors-22-05759-f010:**
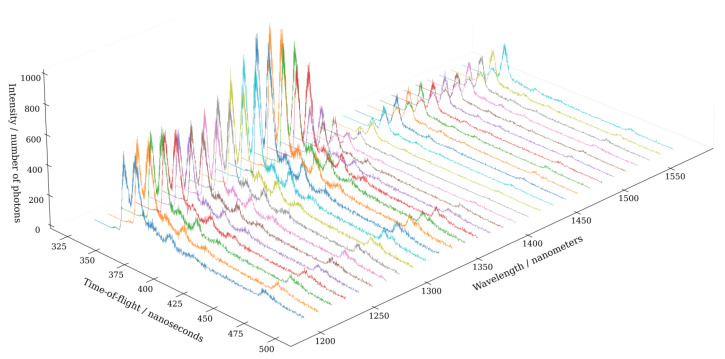
Time-of-flight histogram for each individual wavelength channel. The return waveform from the spruce (*Picea abies*) target shows multiple echoes originating from the needles, pulvinus, branches, and the trunk of the tree. Additionally, the echoes from trees located behind the main target are visible in the data. The intensity has been computed as a sum over 10,000 frames.

**Figure 11 sensors-22-05759-f011:**
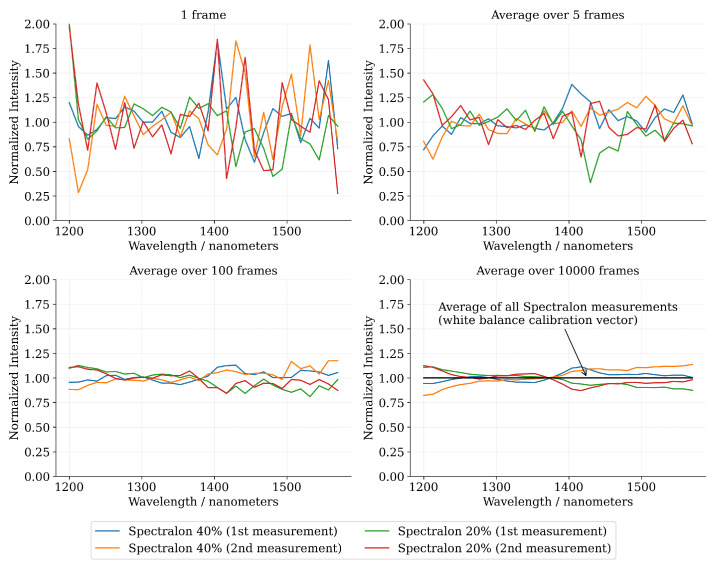
The relative spectral reflectance curves of the Spectralon targets with various block sizes. The white balance calibration vector has been computed as an average over the signal-magnitude normalized Spectralon spectra $S_{n}$ with the frame count set at $N_{frames}$ = 10,000 in order to maximize the signal-to-noise ratio. The bottom-right figure illustrates the average of the four Spectralon spectra as a black curve, which is by definition identity at all wavelength channels (The black curve is equivalent to the relative spectral reflectance curve of the white balance vector).

**Figure 12 sensors-22-05759-f012:**
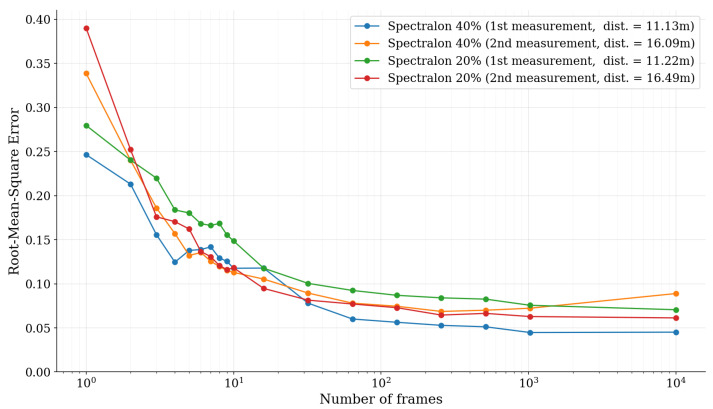
Root-Mean-Square Error (RMSE) between the relative reflectance spectra of Spectralon measurements and an ideal (“flat”) white balance spectrum as a function of the block size $N_{frames}$.

**Figure 13 sensors-22-05759-f013:**
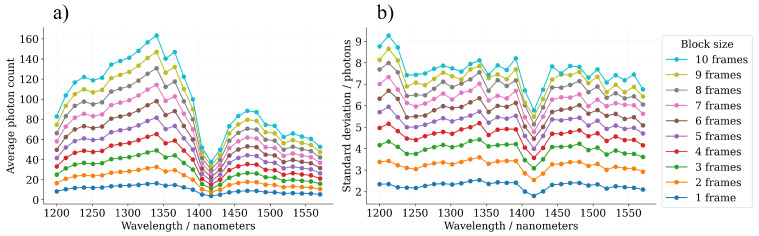
The block-wise (**a**) average photon count and (**b**) photon count standard deviation of the Spectralon 40% target as a function of the block size $N_{frames}\in \left[1,10\right]$.

**Figure 14 sensors-22-05759-f014:**
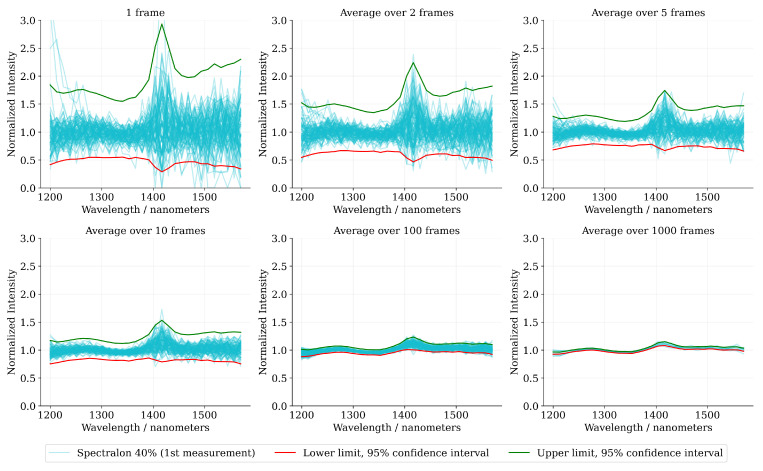
The relative spectral reflectance curves of Spectralon 40% sample, along with the theoretical 95% confidence intervals for various block sizes. Each subplot visualizes 100 sample spectra, except the last one with block size of $N_{frames}=1000$, which visualizes 10 sample spectra. The confidence intervals were calculated by estimating the average photon count over the whole sample sequence of 10,000 consecutive frames.

**Figure 15 sensors-22-05759-f015:**
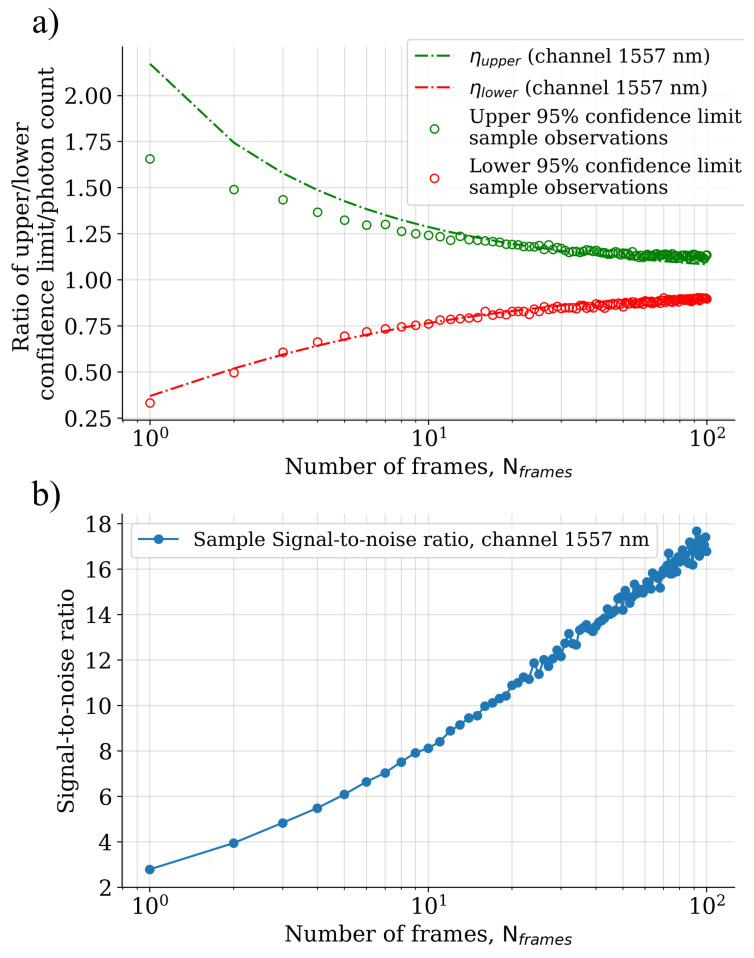
(**a**) The theoretical spectral reflectance measurement accuracy confidence limits $\eta_{\mathrm{lower}}$ and $\eta_{\mathrm{upper}}$, and the relative confidence limits computed from the empirical distribution function with respect to the block size. (**b**) The signal-to-noise ratio as a function of the block size. The observations have been computed from the Spectralon 40% sample. The average photon count for channel λx=1557 nm was approximately E[S(λx)]≈6.04 photons per frame.

**Figure 16 sensors-22-05759-f016:**
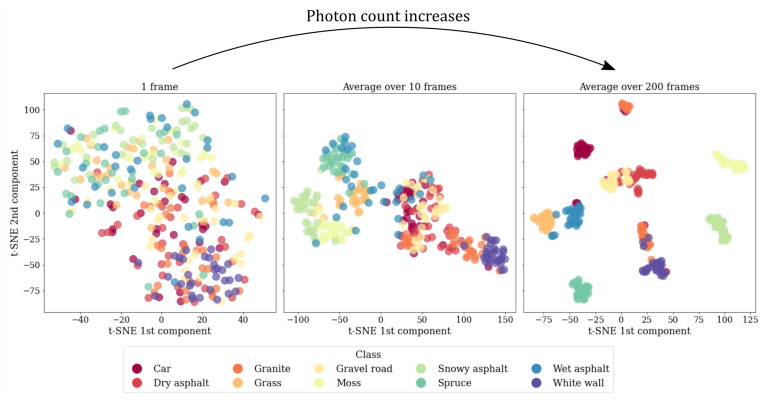
A visualization of the dataset samples that have been embedded in a two-dimensional t-SNE space (perplexity = 5.0). The input data consist of the relative spectral reflectance curves that have been accumulated over a single frame (left-hand side), 10 frames (centre), and 200 frames (right-hand side).

**Figure 17 sensors-22-05759-f017:**
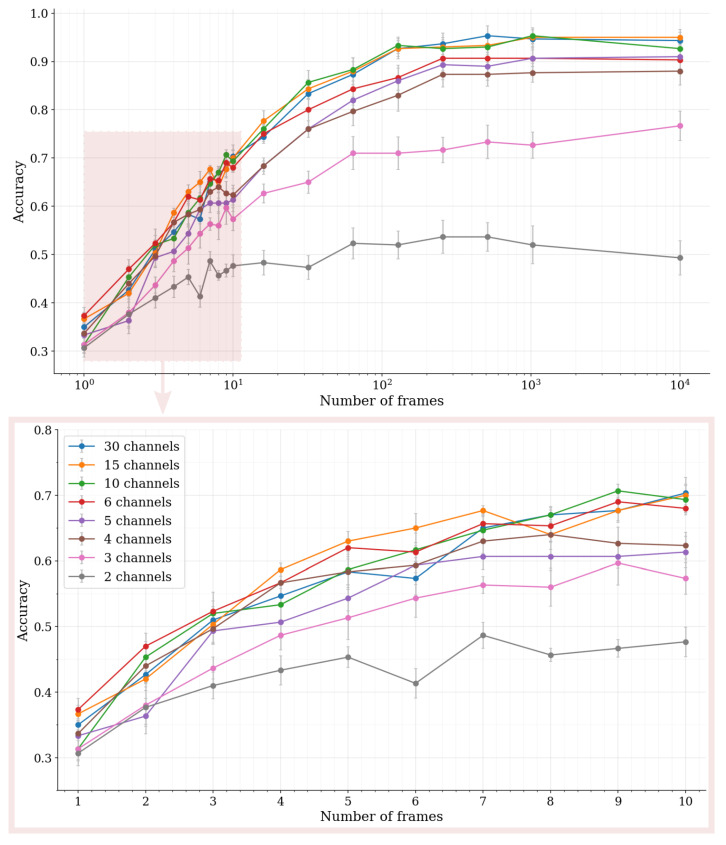
The mean classification accuracy (5-fold cross-validation) in the test set with respect to the block size (number of frames). The error bars denote the standard error of the mean (SEM).

## Data Availability

Not applicable.

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
