# Peer review of "Feasibility of Hyperspectral Single Photon Lidar for Robust Autonomous Vehicle Perception"

_sensors, 2022, doi:10.3390/s22155759_

Round 1

Reviewer 1 Report

This paper shows, in concrete, a suitable design of a NIR spectrophotometer, based on the LIDAR principle. The work is excellent, and I propose to accept publication in Sensors, as written.

My only comment or suggestion to improve the impact is to provide a comparison between the reflectance reported results (obtained with the proposed setup) with reflectance measured with a commercial NIR spectrophotometer. This comparison would have directly shown the effectiveness of the proposed setup.

Author Response

Thank you for dedicating your time and effort to provide valuable feedback on our manuscript. We are grateful for your insightful comments. We have provided response to your comments and suggestions below:

This paper shows, in concrete, a suitable design of a NIR spectrophotometer, based on the LIDAR principle. The work is excellent, and I propose to accept publication in Sensors, as written.

My only comment or suggestion to improve the impact is to provide a comparison between the reflectance reported results (obtained with the proposed setup) with reflectance measured with a commercial NIR spectrophotometer. This comparison would have directly shown the effectiveness of the proposed setup.

Response: Thank you for your suggestion. We have already discussed the possibility of comparing the reflectance measurements from the prototype hyperspectral single photon lidar to a calibrated spectrophotometer both in a theoretical and in a practical way. Unfortunately, the current tight schedule does not permit us to perform the measurements, process the data and to incorporate the results to the submitted manuscript. However, we will definitely add this type of comparison in future publications, hopefully quite soon.

In addition to your proposal, we have also considered examining the effects on the return pulse photon count that arise from the time-filtering step with varying target impulse response function in a separate study. We believe that the results considering the target geometry and the measured (and validated) absolute reflectance could be nicely provided in combination in a further paper due to their close characteristic relationship.

Reviewer 2 Report

The manuscript is well-written and the hyper-spectral system is well-built. However, it is only suitable for single-point LiDAR measurements. Could the authors justice more on how to explore their system for imaging applications?

I  have one comment on the transmitter of the authors' system. In order to enhance the illumination beam uniformity across the wavelengths of interest, is it better to use reflective collimator?

Author Response

Thank you for dedicating your time and effort to provide valuable feedback on our manuscript. We are grateful for your insightful comments. We have provided response to your comments and suggestions below:

The manuscript is well-written and the hyper-spectral system is well-built. However, it is only suitable for single-point LiDAR measurements. Could the authors justice more on how to explore their system for imaging applications?

Response: Thank you for pointing out the practical limitations of the prototype hyper-spectral system. In this manuscript, we preferred to guide the discussion towards the effects that arise from the discrete nature of the extremely weak return pulse photon counts in single photon sensitive lidars and examine how that might affect the reliability of either the spectral measurement or, in the case of monochromatic lidar devices, the reflectance measurement accuracy. Therefore, we have not included measurement results that have been obtained by scanning the illumination beam across the measurement scene. However, we will address your concerns in further studies, hopefully quite soon.

We have already designed and built a rotating mirror type optical head assembly for the prototype hyper-spectral system, but the development work is still under progress. Therefore, we feel that it is probably better to introduce the results from the updated system in a separate paper.

In order to address your comment on this manuscript, we have incorporated changes to section 3.1 (lines 181 - 184 (referring to the indexing at the updated manuscript)) to reflect the absence of scanning mechanism in the current hardware and to clarify the requirements for obtaining scanning ability in future hardware revisions. The changes/additions are as follows: “Beam scanning can be realized on the prototype hyperspectral single photon lidar by employing standard beam scanning mechanisms, such as rotating multi-faceted-, Palmer scanning- or oscillating mirrors. Additionally, the optical head assembly can be designed to be operated on a rotating assembly.”

I  have one comment on the transmitter of the authors' system. In order to enhance the illumination beam uniformity across the wavelengths of interest, is it better to use reflective collimator?

Response: Yes, you are absolutely correct. A reflective collimator would be a better option instead of the current refractive collimator as it would introduce less chromatic aberration to the illumination beam. As was mentioned earlier, we are in the process of further developing the hardware. The updated version of the prototype device should account for the non-idealities in the optical paths, as well as improve upon some of the previous architectural design choices.